# SAM2-3DMED:
# EMPOWERING SAM2 FOR 3D MEDICAL IMAGE SEGMENTATION

## ABSTRACT

Accurate segmentation of 3D medical images is critical for clinical applications like disease assessment and treatment planning. While the Segment Anything Model 2 (SAM2) has shown remarkable success in video object segmentation by leveraging temporal cues, its direct application to 3D medical images faces two fundamental domain gaps: 1) the bidirectional anatomical continuity between slices contrasts sharply with the unidirectional temporal flow in videos, and 2) precise boundary delineation—crucial for morphological analysis—is often underexplored in video tasks. To bridge these gaps, we propose SAM2-3dMed, an adaptation of SAM2 for 3D medical imaging. Our framework introduces two key innovations: 1) a Slice Relative Position Prediction (SRPP) module explicitly models bidirectional inter-slice dependencies by guiding SAM2 to predict the relative positions of different slices in a self-supervised manner; 2) a Boundary Detection (BD) module enhances segmentation accuracy along critical organ and tissue boundaries. Extensive experiments on three diverse medical datasets (the Lung, Spleen, and Pancreas in the Medical Segmentation Decathlon (MSD) dataset) demonstrate that SAM2-3dMed significantly outperforms state-of-the-art methods, achieving superior performance in segmentation overlap and boundary precision. Our approach not only advances 3D medical image segmentation performance but also offers a general paradigm for adapting video-centric foundation models to spatial volumetric data.

## 1 INTRODUCTION

Precise 3D medical image segmentation serves as a cornerstone of modern precision medicine, critically underpinning applications in disease diagnosis, treatment planning (e.g., surgical navigation and radiotherapy targeting), and therapeutic efficacy assessment. Over the past decade, deep learning, particularly Fully Convolutional Network (FCN) architectures like U-Net and its variants, has achieved remarkable success in this domain (Long et al., 2015; Milletari et al., 2016; Çiçek et al., 2016; Isensee et al., 2021; Ronneberger et al., 2015; Hatamizadeh et al., 2022). However, these methods predominantly rely on fully supervised learning paradigms, whose performance is heavily constrained by the availability of large volumes of high-quality, meticulously annotated 3D data with slice-by-slice labels. In clinical practice, pixel-level annotation of 3D volumetric data (e.g., CT or MRI) is an extremely time-consuming, expensive, and highly specialized task, creating a significant "annotation bottleneck" that hinders the widespread clinical deployment of deep learning models (Zheng et al., 2020; Jiao et al., 2024; Liu et al., 2023a).

To overcome this bottleneck, researchers have primarily explored two avenues: Semi-Supervised Learning (SSL) and Transfer Learning (TL) (Cheplygina et al., 2019; Tajbakhsh et al., 2020). SSL aims to leverage unlabeled data to improve model performance, but its effectiveness is closely tied to the quality and distribution of the unlabeled data, and it carries the risk of introducing biases from the unlabeled data (Jiao et al., 2024; Xie et al., 2020). In contrast, TL offers a more promising paradigm: it learns general visual representations from large-scale source domains (e.g., natural images or videos) and transfers them to data-scarce target domains (e.g., medical images), thereby reducing reliance on annotated data from the target domain (Shin et al., 2016; Zhou et al., 2019).

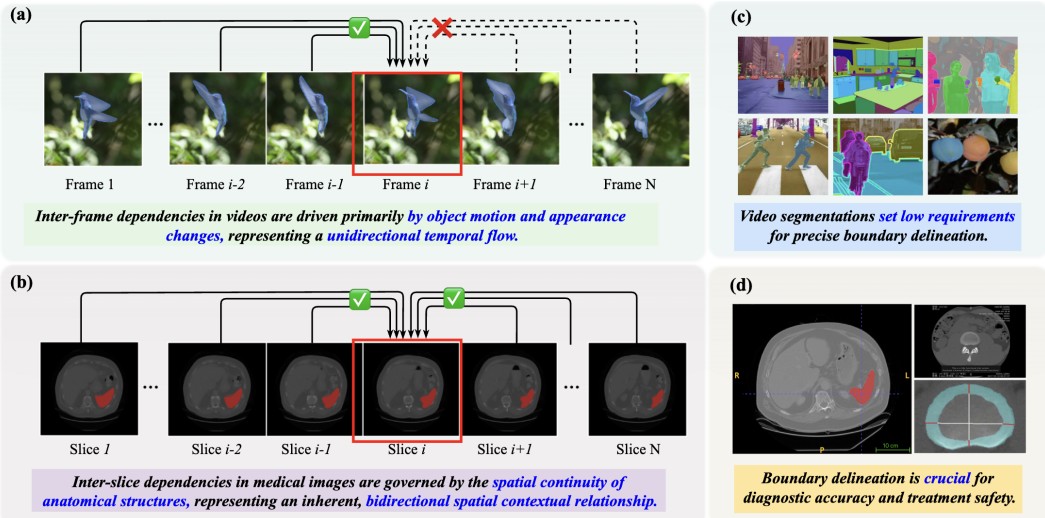

Figure 1: The comparison focuses on inter-frame dependencies in videos (a) vs. inter-slice dependencies in 3D medical images (b), and the importance of boundary segmentation for videos (c) vs. that for medical images (d).

Recently, the rise of foundation models has opened new opportunities for transfer learning. Among them, the Segment Anything Model (SAM) (Kirillov et al., 2023) and its video extension SAM2 (Ravi et al., 2024) have demonstrated unprecedented general segmentation capabilities. Trained on a dataset of over one billion masks, SAM possesses powerful zero-shot generalization abilities. However, directly applying SAM to medical image segmentation faces significant limitations: its training data primarily consists of natural images, lacking the specific textures, morphologies, and pathological features characteristic of medical images, often leading to suboptimal performance on medical data. More critically, SAM and most of its variants (e.g., MedSAM (Ma et al., 2024)) are inherently 2D architectures, incapable of effectively modeling the crucial inter-slice spatial dependencies inherent in 3D medical volumetric data.

The emergence of SAM2 provides a new perspective for 3D medical image segmentation (Kirillov et al., 2023; Ma et al., 2024; Wu et al., 2025). It effectively models temporal dependencies in videos through a memory mechanism and has demonstrated excellent performance in video object segmentation (Ravi et al., 2024; Oh et al., 2019) . Intuitively, a 3D medical volume can be analogized as a "spatial video," where consecutive slices are analogous to video frames (Zhu et al., 2024; Lee et al., 2025). However, this analogy suffers from fundamental domain discrepancies: **1) Nature of Dependencies**: Inter-frame dependencies in videos are primarily driven by object motion and appearance changes, representing a unidirectional temporal flow (Simonyan & Zisserman, 2014) Fig. 1 (a). In contrast, inter-slice dependencies in medical images are governed by the spatial continuity of anatomical structures, representing an inherent, bidirectional spatial contextual relationship (Çiçek et al., 2016; Litjens et al., 2017) Fig. 1 (b). **2) Focus of Task Objectives**: Video segmentation often prioritizes target integrity and temporal consistency, with relatively lower requirements for precise boundary delineation (Azulay et al., 2022; Baghbaderani et al., 2024) Fig. 1 (c). Conversely, medical segmentation demands exact boundary delineation for organs, tumors, and other targets, which is directly crucial for diagnostic accuracy and treatment safety (Pan et al., 2024; Xu et al., 2024) Fig. 1 (d). These discrepancies imply that directly transferring SAM2 to 3D medical image segmentation is suboptimal and may even fail to converge effectively. Therefore, adaptation of SAM2 is essential to bridge the domain gap between videos and medical volumetric data.

To address the above challenges, we propose **SAM2-3dMed**, a novel adaptation framework specifically tailored for 3D medical image segmentation. Our core idea is: while inheriting the powerful feature representation capabilities of SAM2, we introduce novel self-supervised tasks and structural optimizations to guide the model to understand the spatial structural properties of medical volumes and focus on critical boundary information. Specifically, our innovations include: **1) Slice Relative**

**Position Prediction (SRPP) Module**: We design a self-supervised slice relative position prediction task. This module compels the model to predict the relative positions between any two slices within a volume, explicitly guiding the model to learn bidirectional spatial context and anatomical structural continuity between slices, thereby successfully transforming SAM2 from modeling temporal dependencies to modeling spatial dependencies. **2) Boundary Detection (BD) Module**: Given the extreme importance of precise boundary segmentation in clinical practice, we introduce a BD module parallel to the mask decoder. This module focuses on extracting and enhancing boundary features, significantly improving the segmentation accuracy of organ and lesion contours through explicit supervision of boundary prediction, which is vital for subsequent morphological analysis and surgical planning.

We conduct extensive experiments on three public datasets, and the results demonstrate that SAM2-3dMed effectively enhances the performance of 3D medical image segmentation and outperforms existing state-of-the-art methods. The main contributions of this paper are:

1. We systematically analyze the core challenges and domain discrepancies when transferring the video foundation model SAM2 to 3D medical image segmentation.

2. We propose a novel adaptation framework, SAM2-3dMed, which addresses two key issues—spatial dependency modeling and boundary accuracy improvement—through the SRPP and BD innovative modules, respectively.

3. Extensive experiments validate the effectiveness of our method, providing new insights and a strong baseline for future exploration of video foundation models in medical imaging.

## 2 RELATED WORKS

### 2.1 3D MEDICAL IMAGE SEGMENTATION

The pursuit of accurate 3D medical image segmentation is fundamental to quantitative medical image analysis, enabling precise delineation of anatomical structures and pathological regions for diagnosis, treatment planning, and disease monitoring. Early foundational works largely relied on fully supervised learning paradigms, with architectures like 3D U-Net (Çiçek et al., 2016) and its variants (e.g., V-Net (Milletari et al., 2016)) establishing themselves as powerful benchmarks. These models excel by learning hierarchical features directly from volumetric data, capturing essential intra-slice and inter-slice context. The field further advanced with the introduction of transformer-based architectures like UNETR (Hatamizadeh et al., 2022), which aimed to capture long-range spatial dependencies within 3D volumes through self-attention mechanisms. Frameworks such as nnU-Net (Isensee et al., 2021) have demonstrated remarkable performance by intelligently automating network design and preprocessing, highlighting the maturity of fully supervised methods on datasets with abundant annotations.

However, the performance of these fully supervised methods is intrinsically bounded by the "annotation bottleneck" — the prohibitive cost, time, and expertise required to acquire dense, high-quality voxel-level labels for 3D medical images (Zheng et al., 2020; Jiao et al., 2024; Liu et al., 2023a). This limitation has catalyzed a strategic shift in research focus towards data-efficient learning paradigms, primarily Semi-Supervised Learning (SSL) and Transfer Learning (TL), which aim to maximize performance with minimal annotated data (Jiao et al., 2024; Kim et al., 2022).

### 2.2 DATA-EFFICIENT LEARNING

Semi-Supervised Learning (SSL) strategies leverage abundant unlabeled data through methods like consistency regularization and pseudo-labeling. While effective, their efficacy is inherently constrained by the quality and distribution of the in-domain unlabeled data. Performance can be sensitive to noise propagation from inaccurate pseudo-labels, and biases present in the unlabeled set may be amplified, potentially limiting generalization across different domains (Arazo et al., 2020; Ouali et al., 2020).

In contrast, Transfer Learning (TL) aims to overcome data scarcity by adapting rich visual representations from models pre-trained on large-scale source domains. This paradigm has proven highly successful for 2D medical image segmentation. However, its application to 3D volumetric analysis

has been historically hindered by a critical gap: the absence of powerful, pre-trained 3D foundation models. This scarcity has typically forced 3D models to be trained from scratch, leaving them heavily dependent on large annotated datasets (Zhou et al., 2019; Çiçek et al., 2016).

## 2.3 THE RISE OF FOUNDATION MODELS

Foundation models like the Segment Anything Model (SAM) (Kirillov et al., 2023) have revolutionized 2D segmentation. While adaptations like MedSAM (Ma et al., 2024) show success, they inherit SAM's core limitation: processing 3D volumes as isolated 2D slices, thus ignoring volumetric context. The subsequent SAM2 (Ravi et al., 2024) addresses sequential data by modeling inter-frame dependencies in video. This inspired the "spatial video" paradigm, where SAM2 is seen as a natural backbone for 3D medical segmentation like MedSAM-2 (Wang et al., 2024; Zhu et al., 2024).

However, early explorations reveal a critical challenge: naively applying SAM2 to medical volumes can yield suboptimal results, sometimes underperforming even fine-tuned 2D models (Dong et al., 2024; Zhu et al., 2024). This performance gap indicates that the analogy between temporal video frames and spatial medical slices is powerful but incomplete, necessitating specialized adaptations to bridge this domain-specific gap.

## 3 METHODOLOGY

### 3.1 PROBLEM FORMULATION & OVERVIEW

Let a 3D medical volume be represented as $X \in \mathbb{R}^{3 \times D \times H \times W}$, where $D$ denotes the number of slices along the depth dimension, and $H$ and $W$ represent the spatial dimensions of each slice. The corresponding ground-truth segmentation mask is denoted as $GT_{seg} \in \{0, 1\}^{K \times D \times H \times W}$, where $K$ is the number of target classes (e.g., organs, tissues). Our goal is to learn a mapping function $F_\theta$, parameterized by $\theta$, that produces a precise segmentation mask $P_{seg}$, while addressing two domain-specific challenges in 3D medical image segmentation: **1) Bidirectional Spatial Dependency**: Modeling anatomical continuity across slices, which is inherently bidirectional (unlike unidirectional temporal dependencies in videos). **2) Boundary Delineation Precision**: Ensuring high accuracy in segmenting critical boundaries like organ contours and lesion margins. Our proposed framework, SAM2-3dMed, adapts the SAM2 architecture for 3D medical image segmentation by introducing two novel modules that address the above challenges while leveraging SAM2's powerful feature representation capabilities. The overall framework consists of the following components (Fig. 2):

- **SAM2 Backbone**: We utilize SAM2's pre-trained *Image Encoder* to extract features from input volumetric data. The encoder processes each slice independently, generating a feature tensor $Z \in \mathbb{R}^{C \times D \times H' \times W'}$. To preserve SAM2's generalization ability, we keep the encoder frozen during training.

- **SAM2-based Segmentation Module**: This module comprises SAM2's *Memory Attention* and *Mask Decoder* components. It takes $Z$ as input and produces the segmentation map $P_{seg} \in \mathbb{R}^{K \times D \times H \times W}$. The module is optimized using a segmentation loss $\mathcal{L}_{\text{seg}}$ (e.g., Dice loss) that measures the discrepancy between $P_{seg}$ and $GT_{seg}$.

- **Slice Relative Position Prediction (SRPP) Module**: To explicitly model bidirectional inter-slice dependencies, we introduce a self-supervised auxiliary task. This module consists of a Transformer-based *Slice Relative Position Encoder* that captures global context across slices and a lightweight MLP-based *Slice Relative Position Predictor*. The module is optimized through minimizing the slice relative position prediction loss $\mathcal{L}_{\text{srpp}}$ (e.g., MSE loss) between the predicted $P_{pos} \in \mathbb{R}^{D \times D}$ and actual relative positions $GT_{pos} \in \{-D+1, -D+2, \ldots, 0, \ldots, D-2, D-1\}^{D \times D}$.

- **Boundary Detection (BD) Module**: To enhance boundary segmentation precision, we incorporate a parallel branch that focuses on boundary-aware feature learning. This module uses a structure similar to SAM2's *Memory Attention* and *Mask Decoder* and operates on the features from the encoder ($Z$). It outputs a predicted boundary map $P_{bd} \in \mathbb{R}^{K \times D \times H \times W}$, which is supervised via a boundary detection loss $\mathcal{L}_{\text{bd}}$ (e.g.,

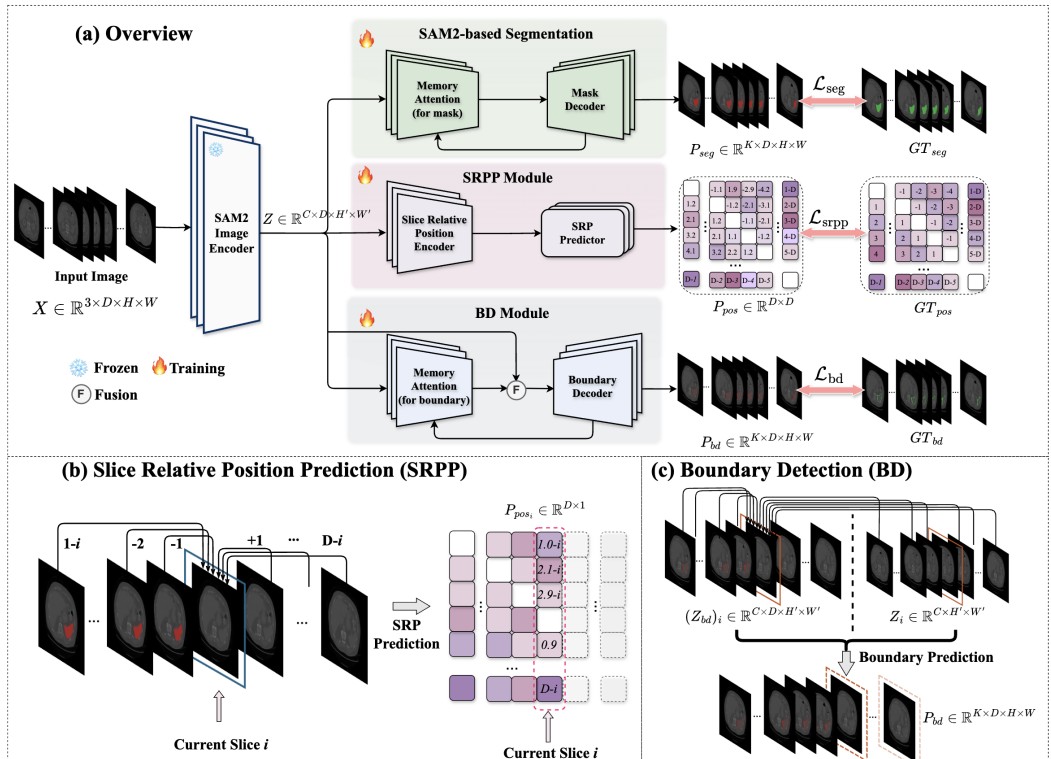

Figure 2: Overview of the proposed SAM2-3dMed Model.

weighted binary cross-entropy) against the ground-truth boundaries derived from $GT_{seg}$ ($GT_{bd} \in \{0, 1\}^{K \times D \times H \times W}$).

During the training phase, the pre-trained *Image Encoder* of SAM2 is kept frozen. Through minimizing the combination of the three loss items $\mathcal{L}_{seg}$, $\mathcal{L}_{srpp}$, and $\mathcal{L}_{bd}$, the *Memory Attention* and *Mask Decoder* blocks in the SAM2-based Segmentation module are fine-tuned with the help of the SRPP and BD modules. This strategy ensures efficient training and mitigates overfitting, especially when annotated medical data is limited.

## 3.2 SLICE RELATIVE POSITION PREDICTION (SRPP) MODULE

Accurate modeling of inter-slice dependencies is critical for 3D medical image analysis, as anatomical structures exhibit continuous spatial relationships across slices—a characteristic fundamentally distinct from the unidirectional temporal flow in videos. While self-supervised pretext tasks like temporal order prediction have proven effective for learning representations in videos (Hu et al., 2021), we adapt this concept to capture bidirectional spatial context in volumetric medical data. The SRPP module is designed to explicitly guide the model in learning the inherent anatomical continuity between slices in a self-supervised manner, thereby enhancing the spatial awareness of the SAM2 backbone for 3D segmentation.

The SRPP module takes the feature tensor $Z \in \mathbb{R}^{C \times D \times H' \times W'}$ extracted by SAM2's frozen *Image Encoder* as input and processes $Z$ in a slice-wise manner. For each slice (e.g., Slice-$i$), the module predicts the position of any other slice (e.g., Slice-$j$) relative to it as follows:

$$(P_{\text{pos}})_{i,j} = \text{SRPP}(Z_i, Z_j) \tag{1}$$

where $Z_i$ and $Z_j \in \mathbb{R}^{C \times H' \times W'}$ are feature representations of the $i$-th and $j$-th slices, and $(P_{\text{pos}})_{i,j}$ denotes the predicted relative offset. The ground truth is defined as $(GT_{\text{pos}})_{i,j} = j - i$. The SRPP

function is implemented via a lightweight Transformer-based encoder for capturing global slice interactions, followed by an MLP predictor for regression. The module is optimized using a mean squared error (MSE) loss aggregated over all unique slice pairs:

$$\mathcal{L}_{\text{srpp}} = \frac{\sum_{i=1}^{D} \sum_{j=1, j \neq i}^{D} \left( (P_{\text{pos}})_{i,j} - (GT_{\text{pos}})_{i,j} \right)^2}{D(D-1)} \tag{2}$$

This loss is jointly minimized with the segmentation and boundary losses during training, ensuring the backbone features encode rich spatial structural priors. The SRPP module requires no additional annotations and operates purely in a self-supervised manner. By forcing the model to reason about slice ordering, it enhances the ability of SAM2-based Segmentation module to capture bidirectional anatomical continuity, addressing a key domain gap between video and medical data.

### 3.3 Boundary Detection (BD) Module

Precise boundary delineation is critical in medical image segmentation. SAM2, however, lacks explicit mechanisms for boundary precision. To address this, we introduce a dedicated BD Module, which enhances SAM2's segmentation decoder with boundary-aware feature learning and cross-attention mechanisms, ensuring high fidelity along critical edges.

The BD Module shares the core structure of SAM2's *Memory Attention* and *Mask Decoder* but optimized using the boundary detection loss. The module also processes the feature tensor $Z \in \mathbb{R}^{C \times D \times H' \times W'}$ from SAM2's frozen *Image Encoder* in a slice-wise manner. For each slice (e.g., Slice-$i$), the *Memory Attention (for boundary)* block firsts extracts the boundary aware features $(Z_{bd})_i \in \mathbb{R}^{C \times D \times H' \times W'}$. The module then fuses the feature representation for the slice and the boundary $Z_i \in \mathbb{R}^{C \times H' \times W'}$ and $(Z_{bd})_i$ via a cross-attention mechanism to refine boundary localization:

$$(Z'_{\text{bd}})_i = \text{softmax} \left( \frac{(Q_{\text{bd}})_i K_i^T}{\sqrt{d_K}} \right) V_i \tag{3}$$

where $(Q_{\text{bd}})_i$, $K_i$, and $V_i$ are linear projections of $(Z_{bd})_i$, $Z_i$, and $Z_i$, respectively.

The *Memory Attention(for boundary)* block enriches $(Z_{bd})_i$ with information from its "previous" slices, and this cross-attention mechanism further allows boundary queries to attend to inter-slice dependencies, thereby enhancing spatial coherence. The refined boundary features are further processed as:

$$(Z_{\text{bd}}^{\text{out}})_i = \text{MLP}(\text{LayerNorm}((Z'_{\text{bd}})_i)) + (Z'_{\text{bd}})_i \tag{4}$$

ensuring stability and feature richness through residual connections. The final boundary-aware segmentation logits are derived from the boundary-aware features of all slices $Z_{\text{bd}}^{\text{out}} \in \mathbb{R}^{C \times D \times H' \times W'}$.

Boundary pixels are inherently sparse, leading to severe class imbalance. We mitigate this using a weighted binary cross-entropy loss $\mathcal{L}_{\text{bd}}$, which amplifies gradients for boundary pixels:

$$\mathcal{L}_{\text{bd}} = \frac{N_{\text{non-bd}}}{N} \sum_{j \in \Omega_{\text{bd}}} \text{BCE} \left( (P_{\text{bd}})_j, (GT_{\text{bd}})_j \right) + \frac{N_{\text{bd}}}{N} \sum_{j \in \Omega_{\text{non-bd}}} \text{BCE} \left( (P_{\text{bd}})_j, (GT_{\text{bd}})_j \right) \tag{5}$$

where $N_{\text{bd}}$ and $N_{\text{non-bd}}$ denote the number of boundary and non-boundary pixels, respectively. This weighting scheme ensures the model prioritizes accurate boundary prediction without neglecting non-boundary regions.

### 3.4 Loss Functions

SAM2-3dMed is optimized through minimizing a combination of the object segmentation loss $\mathcal{L}_{\text{seg}}$, the slice relative position prediction loss $\mathcal{L}_{\text{srpp}}$, and the boundary detection loss $\mathcal{L}_{\text{bd}}$. Specifically, the total loss for SAM2-3dMed is defined as follows:

$$\mathcal{L}_{\text{total}} = \mathcal{L}_{\text{seg}} + \lambda_1 \mathcal{L}_{\text{srpp}} + \lambda_2 \mathcal{L}_{\text{bd}} \tag{6}$$

where $\lambda_1$, and $\lambda_2$ are hyper-parameters for balancing the loss items. Here the object segmentation loss $\mathcal{L}_{\text{seg}}$ is defined as the binary cross entropy loss between the predicted and actual segmentations ($P_{seg} \in \mathbb{R}^{K \times D \times H \times W}$ vs. $GT_{seg} \in \{0,1\}^{K \times D \times H \times W}$).

# 4 EXPERIMENTS

In this section, we conduct extensive experiments to validate the effectiveness of our proposed SAM2-3dMed framework for 3D medical image segmentation. We compare our method against supervised models, SAM-based approaches, and transfer learning baselines. Ablation studies are performed to analyze the contributions of the SRPP module, BD module, and the impact of pre-training.

## 4.1 DATASETS

We evaluate our method on the Medical Segmentation Decathlon (MSD) (Antonelli et al., 2022), a large-scale public dataset for 3D medical image segmentation. The dataset contains diverse, fully anonymized medical imaging cases with expert-verified annotations. We focus on three challenging datasets:

- Lung Tumor: 64 thoracic CT scans with voxel-wise annotations of cancerous nodules.
- Spleen: 41 abdominal CT scans from Memorial Sloan Kettering Cancer Center, with full spleen annotations.
- Pancreas: 281 abdominal CT volumes featuring high variability in tumor morphology and location.

**Baselines:** To evaluate the effectiveness of our method, we compare it against six state-of-the-art medical image segmentation algorithms. This includes strong convolutional neural network (CNN)-based baselines like nnUNet (Isensee et al., 2021); Transformer-based models such as nnFormer (Zhou et al., 2021), MedNeXt(Roy et al., 2023). Besides that, we include Universal-Model (Liu et al., 2023b), MedSAM-2 (Zhu et al., 2024) and a directly fine-tuned SAM2 model (Ravi et al., 2024).

**Evaluation Metrics:** Four metrics are used to evaluate the performance of our model and all baselines. Among these, the Dice similarity coefficient and Intersection over Union (IoU) measure the volumetric overlap between the predicted segmentation and the ground-truth mask. For boundary assessment, we use the 95th percentile of the Hausdorff Distance (HD95) and the Normalized Surface Distance (NSD).

HD95 calculates the 95th percentile of distances between the predicted and ground-truth surfaces, offering a measure of boundary deviation that is robust to outliers. The NSD is defined as the percentage of surface voxels that are within an acceptable tolerance distance. For Dice, IoU, and NSD, higher values indicate better performance, whereas a lower value is desirable for HD95. NSD is less sensitive to minor disagreements in large volumes compared to Dice.

## 4.2 MAIN RESULTS

To rigorously evaluate the efficacy and generalization capability of our proposed framework, we conducted comprehensive comparative analyses against a suite of state-of-the-art segmentation methods. Quantitative results across all three tasks (Lung, Spleen, Pancreas) are detailed in Table 1. Our method demonstrates superior performance across the three benchmarks, achieving the highest scores in most key metrics, and shows particularly strong gains on the challenging Pancreas segmentation task. As quantified in the last four columns of Table 1, our method outperforms all baselines, improving upon the second-best method by +2.98% in Dice (from 0.6741 to 0.7039), +1.31% in IoU (from 0.5575 to 0.5706), and +3.00% in NSD (from 0.5896 to 0.6196). These consistent improvements across both volumetric overlap (Dice, IoU) and boundary accuracy metrics (HD95, NSD) underscore the effectiveness of our design in addressing the core challenges of 3D medical image segmentation.

As illustrated in Fig. 3, our method generates segmentations with superior structural consistency and boundary precision compared to all baseline methods. Specifically, segmentations from our model exhibit greater anatomical plausibility and fewer inter-slice inconsistencies(highlighted by cyan boxes), validating that the SRPP module successfully guides the model toward learning spatially dependent features across slices. Moreover, the segment boundaries align more closely with

the ground-truth contours(highlighted by orange arrow), confirming the BD module's efficacy in enhancing edge-aware feature learning and yielding morphologically precise outputs. The convergence of quantitative and qualitative evidence positions our framework as a robust, generalizable solution for high-precision volumetric medical image analysis.

Table 1: Comparison with state-of-the-art methods on the three datasets. Best results are highlighted in **bold**.

| Method | Lung | | | | Spleen | | | | Pancreas | | | |
|---|---|---|---|---|---|---|---|---|---|---|---|---|
| | Dice ↑ | IoU ↑ | HD95 ↓ | NSD ↑ | Dice ↑ | IoU ↑ | HD95 ↓ | NSD ↑ | Dice ↑ | IoU ↑ | HD95 ↓ | NSD ↑ |
| SAM2(Fine-tuned) | 0.7266 | 0.6225 | 8.9635 | 0.7919 | 0.9537 | 0.9146 | 8.4642 | 0.9378 | 0.6677 | 0.5374 | 20.8964 | 0.5896 |
| MedSAM-2 | 0.7190 | 0.6288 | 18.3386 | 0.7789 | 0.9418 | 0.8916 | 3.5772 | 0.9151 | 0.5368 | 0.4831 | 39.1401 | 0.4736 |
| nnUNet | 0.7437 | 0.6277 | 8.6284 | 0.7970 | 0.9678 | 0.9381 | 3.6847 | 0.9383 | 0.6518 | 0.5370 | 9.1586 | 0.5732 |
| Universal-Model | 0.7532 | 0.6531 | 20.7765 | 0.8024 | 0.9584 | 0.9203 | 1.8566 | 0.9586 | 0.5988 | 0.4887 | 25.9004 | 0.5186 |
| nnFormer | 0.7508 | 0.6061 | 6.7850 | 0.7437 | 0.9658 | 0.9341 | 1.8323 | 0.9573 | 0.6680 | 0.5575 | **7.8085** | 0.5840 |
| MedNeXt | 0.7243 | 0.6074 | 10.3022 | 0.7413 | 0.9680 | 0.9381 | 1.7848 | 0.9628 | 0.6741 | 0.5519 | 9.8482 | 0.5878 |
| **Ours** | **0.7627** | **0.6544** | **3.5148** | **0.8197** | **0.9727** | **0.9471** | **1.6240** | **0.9742** | **0.7039** | **0.5706** | 14.9232 | **0.6196** |

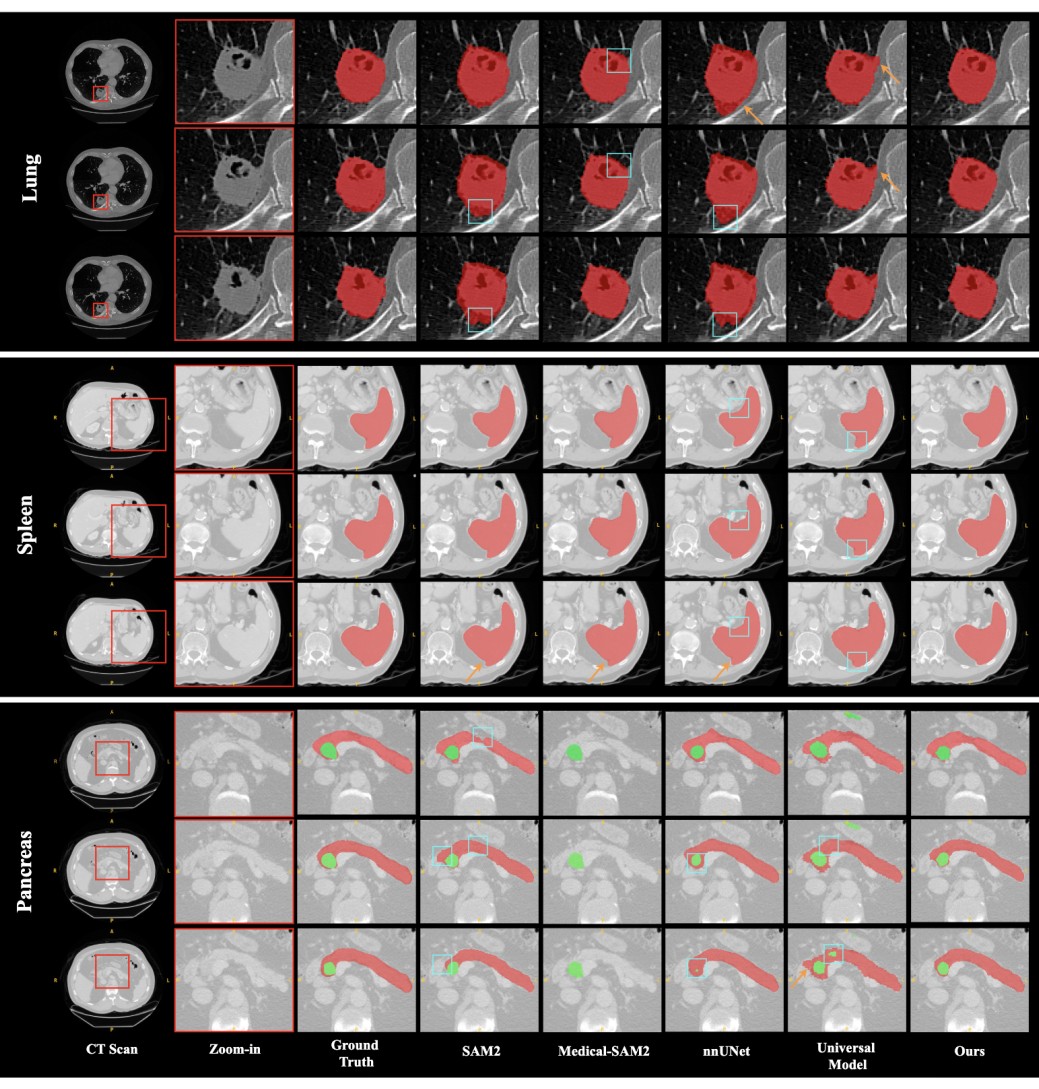

Figure 3: Typical segmentation maps for the three tasks. The cyan boxes highlight lower inter-slice continuity, and the orange arrows highlight worse boundary segmentations.

### 4.3 ABLATION STUDIES

To rigorously evaluate the contribution of each core component in our proposed SAM2-3dMed framework, we conduct systematic ablation experiments. All studies are performed on three datasets under identical training settings and hyperparameters. We design three experimental configurations: 1) w/o Pre-train: Training SAM2-3dMed from scratch, without leveraging the pre-trained SAM2 *Image Encoder* and *Memory Attention* modules; 2) w/o SRPP: Removing the *Slice Relative Position Prediction* module; 3) w/o BD: Removing the *Boundary Detection* module.

As summarized in Table 2, the ablated models exhibit significant performance degradation compared to the full SAM2-3dMed, confirming the necessity of each component.

Table 2: Ablation study of our proposed modules on the three datasets. Best results are highlighted in **bold**.

| Method | Lung | | | | Spleen | | | | Pancreas | | | |
|---|---|---|---|---|---|---|---|---|---|---|---|---|
| | Dice ↑ | IoU ↑ | HD95 ↓ | NSD ↑ | Dice ↑ | IoU ↑ | HD95 ↓ | NSD ↑ | Dice ↑ | IoU ↑ | HD95 ↓ | NSD ↑ |
| w/o Pre-train | 0.6003 | 0.4838 | 9.5642 | 0.6209 | 0.8913 | 0.8144 | 7.8228 | 0.7880 | 0.4353 | 0.3024 | 47.9971 | 0.3619 |
| w/o SRPP | 0.7535 | 0.6408 | 8.5837 | 0.8021 | 0.9672 | 0.9371 | 2.1951 | 0.9589 | 0.6459 | 0.5106 | 21.9181 | 0.5417 |
| w/o BD | 0.7366 | 0.6186 | 9.8799 | 0.7836 | 0.9678 | 0.9379 | 1.9067 | 0.9587 | 0.6727 | 0.5406 | 18.7106 | 0.5897 |
| **Ours** | **0.7627** | **0.6544** | **3.5148** | **0.8197** | **0.9727** | **0.9471** | **1.6240** | **0.9742** | **0.7039** | **0.5706** | **14.9232** | **0.6196** |

**Impact of Transfer Learning**: The w/o Pre-train configuration performs the worst across all metrics and tasks. Compared to our full model, its removal leads to an average Dice score drop of 0.1708 and an average NSD drop of 0.2142 across the three datasets. Specifically, the Dice score decreases by 0.1624 on Lung, 0.0814 on Spleen, and a significant 0.2686 on Pancreas. This underscores the critical role of transfer learning in leveraging pre-trained representations for 3D medical segmentation, especially when annotated data is limited.

**Role of SRPP Module**: Removing the SRPP module causes notable performance declines, with Dice scores dropping by 0.0092 on the Lung dataset and a more substantial 0.0580 on the Pancreas dataset. This decrease highlights the SRPP's efficacy in modeling bidirectional inter-slice dependencies, which is particularly crucial for organs with complex spatial continuity like the pancreas.

**Role of BD Module**: The absence of the BD module leads to a substantial degradation in boundary quality, directly impacting metrics sensitive to edge accuracy. This is evidenced by a drop in NSD of 0.0155 on the Spleen dataset and a considerable increase in the HD95 boundary distance across all datasets. For instance, on the challenging Pancreas task, HD95 worsens from 14.9232 to 18.7106 (an increase of 3.7874). This result underscores the BD module's critical role in ensuring accurate boundary delineation. Notably, its removal also results in significant reductions in volumetric overlap, reducing the Dice score by 0.0312 and IoU by 0.0300 on the Pancreas dataset, which aligns with previous reports that highlight the importance of auxiliary boundary-related tasks.

## 5 CONCLUSION

In this study, we adapted SAM2, known for its powerful capabilities in video object segmentation, for 3D medical image segmentation by leveraging the intrinsic similarities between inter-frame dependencies in videos and inter-slice dependencies in 3D volumes. Our approach addresses two critical domain adaptation challenges through innovative modules: the Slice Relative Position Prediction (SRPP) module enhances SAM2's ability to capture bidirectional inter-slice dependencies in 3D volumes by predicting relative slice positions in a self-supervised manner, while the Boundary Detection (BD) module improves boundary delineation accuracy through an auxiliary boundary segmentation task. Together, these components enable effective transfer of video-based foundation models to medical imaging analysis. The proposed SAM2-3dMed framework represents a promising solution for 3D medical image segmentation, particularly in scenarios where annotated slices are scarce. By explicitly modeling spatial continuity and enhancing boundary precision, our method not only advances segmentation performance but also provides a generalizable paradigm for adapting temporal models to spatial volumetric data across various medical applications.

## 6 ETHICS STATEMENT

We have read and adhered to the ICLR Code of Ethics. This work focuses on medical image segmentation and utilizes public datasets. We do not foresee any direct negative societal impacts or ethical concerns arising from this research.

## 7 REPRODUCIBILITY STATEMENT

To ensure reproducibility, we provide our source code in the supplementary material. Experimental details, hyperparameters, and data processing steps are documented in the appendix.

## 8 THE USE OF LARGE LANGUAGE MODELS (LLMS)

We used LLMs as a writing assistant to improve grammar and clarity. The authors retain full responsibility for all content and scientific claims in this paper.

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

## A  DATASETS DETAILS

The experiments in this study were conducted on three distinct datasets from the Medical Segmentation Decathlon (MSD) challenge: Lung, Spleen, and Pancreas (Antonelli et al., 2022). The MSD is a comprehensive, open-source benchmark designed to evaluate the generalization capabilities of image segmentation algorithms across a variety of medical imaging tasks. It encompasses a wide range of challenges, including small datasets, unbalanced labels, large-ranging object scales, multi-class labels, and multimodal imaging. The data is provided in the Neuroimaging Informatics Technology Initiative (NIfTI) format.

### A.1  LUNG (TASK06)

The lung dataset is focused on the segmentation of lung tumors from computed tomography (CT) scans. This task presents the specific challenge of identifying small cancerous regions within a large imaging volume.

- **Data Source:** The images were sourced from The Cancer Imaging Archive.

- **Modality:** The dataset consists of thoracic CT scans.

- **Composition:** For our experiments, we utilized the original 64 training cases, subdividing them into an 80% split for training and a 20% split for testing.

- **Annotations:** The ground truth labels provide segmentation masks for lung tumors. The primary challenge lies in the segmentation of a small target (the tumor) within a large anatomical context (the lung).

### A.2  SPLEEN (TASK 09)

This dataset is designed for the segmentation of the spleen from abdominal CT scans. The primary imaging modality is contrast-enhanced CT.

- **Data Source:** The data was provided by the Memorial Sloan Kettering Cancer Center and consists of scans from patients undergoing chemotherapy for liver metastases.

- **Modality:** The images are portal-venous phase contrast-enhanced abdominal CT scans.

- Composition: All experiments reported in this paper were conducted using the 41 training cases, which we partitioned into an 80% training set and a 20% test set.

- **Annotations:** Each scan is accompanied by a manually created segmentation of the spleen.

### A.3  PANCREAS (TASK 07)

The pancreas dataset targets the segmentation of the pancreas and any associated tumors from abdominal CT scans. This task is considered particularly challenging due to several factors inherent to the data.

- **Data Source:** This dataset was also contributed by the Memorial Sloan Kettering Cancer Center.

- **Modality:** The images are portal venous phase contrast-enhanced 3D abdominal CT scans.

- **Composition:** In our work, we used the 281 training volumes, creating our own split of 80% for training and 20% for testing.

- **Annotations:** The ground truth masks have separate labels for the pancreas and pancreatic tumors. Key challenges include the significant variability in pancreas shape, blurry boundaries with adjacent organs, and the subtle appearance of tumors. An additional difficulty is the label imbalance between the background, the pancreas, and the typically small tumor structures.

## B TRAINING DETAILS

**Training recipe.** Our model is trained using the AdamW optimizer for a maximum of 500 epochs, employing an early stopping mechanism based on validation performance to prevent overfitting. A key aspect of our training strategy is that the Hiera image encoder backbone, initialized from the SAM2 checkpoint, remains frozen throughout training.

The learning rate for all trainable parameters is initialized to 5.0e-5 and follows a cosine annealing schedule, decaying to one-tenth of its initial value over the course of the training. We apply a weight decay of 0.1 to all trainable parameters. The model is trained on 4 NVIDIA RTX 4090 GPUs with a batch size of 4.

**Data augmentation.** We apply a comprehensive set of data augmentation techniques during training to improve the model's robustness and generalization. For each sequence, the following augmentations are applied consistently across all frames. First, we apply geometric transformations including random horizontal flipping and random affine transformations, which consist of rotations in the range of [-25°, 25°] and shear transformations up to 20°. Following this, all frames are resized to a fixed resolution of 512x512 pixels.

To further augment the data, we introduce appearance perturbations through random Gaussian noise and random Gaussian blur. Additionally, we employ a RandomMosaic augmentation with a probability of 0.15.

## C EVALUATION METRICS

We evaluate the segmentation performance using four standard metrics that assess both regional overlap and boundary accuracy. Let $P$ denote the set of predicted voxels and $G$ be the ground truth.

**Dice Similarity Coefficient (Dice)** and **Intersection over Union (IoU)** are used to measure regional overlap. They are defined as:

$$\text{Dice}(P, G) = \frac{2 \times |P \cap G|}{|P| + |G|} \tag{7}$$

$$\text{IoU}(P, G) = \frac{|P \cap G|}{|P \cup G|} \tag{8}$$

where $|\cdot|$ represents the number of voxels in a set. Both metrics range from 0 to 1, with higher values indicating better overlap.

For boundary-based evaluation, we use the **95% Hausdorff Distance (HD95)** and **Normalized Surface Distance (NSD)**. Let $S_P$ and $S_G$ be the sets of surface points for the prediction and ground truth, respectively. The HD95 measures the 95th percentile of distances between the surfaces, providing a robust metric for boundary discrepancies.

$$\text{HD95}(P, G) = \max\left(K_{p \in S_P}^{95th}\left(\min_{g \in S_G} \|p - g\|\right), K_{g \in S_G}^{95th}\left(\min_{p \in S_P} \|g - p\|\right)\right) \tag{9}$$

where $K^{95th}$ is the 95th percentile operator. A lower value is better. The NSD calculates the fraction of surface points within a specified tolerance $\tau$, which is often more clinically relevant.

$$\text{NSD}(P, G; \tau) = \frac{1}{|S_P| + |S_G|}\left(\sum_{p \in S_P} \mathbb{I}\left(\min_{g \in S_G} \|p - g\| \leq \tau\right) + \sum_{g \in S_G} \mathbb{I}\left(\min_{p \in S_P} \|g - p\| \leq \tau\right)\right) \tag{10}$$

where $\mathbb{I}(\cdot)$ is the indicator function. A higher NSD value indicates better boundary agreement.

## D DETAILED ABLATION STUDIES AND ANALYSIS

### D.1 EFFECTIVENESS OF TRANSFER LEARNING

To validate the effectiveness of leveraging pre-trained weights for initializing our model, we conducted a comparative experiment. We compared our full model, which utilizes weights pre-trained

on a large-scale natural image dataset from original SAM2, against an identical model trained from scratch ("w/o Pre-train"). The results of this comparison and visualization are detailed in Table 3 and Fig. 4.

The empirical results clearly demonstrate that pre-training brings substantial performance improvements across all three datasets and all evaluation metrics. For instance, on the challenging Pancreas dataset, employing pre-training boosts the Dice score from 0.4353 to 0.7039 (a 61.7% relative improvement) and reduces the 95th percentile Hausdorff Distance (HD95) dramatically from 47.9971 to 14.9232. Similar significant gains are observed on the Lung dataset, where the Dice score increases from 0.6003 to 0.7627, and on the Spleen dataset, which sees a rise from 0.8913 to 0.9727.

This consistent and significant performance enhancement underscores the crucial role of transfer learning in our approach. The features learned from large-scale datasets provide a superior initialization for the network's weights, enabling the model to converge to a much better solution and achieve higher segmentation accuracy, especially in medical imaging scenarios where annotated data is often scarce.

Table 3: Ablation study of Pre-train on the three datasets. Best results are highlighted in **bold**.

| Method | Lung | | | | Spleen | | | | Pancreas | | | |
|---|---|---|---|---|---|---|---|---|---|---|---|---|
| | Dice ↑ | IoU ↑ | HD95 ↓ | NSD ↑ | Dice ↑ | IoU ↑ | HD95 ↓ | NSD ↑ | Dice ↑ | IoU ↑ | HD95 ↓ | NSD ↑ |
| w/o Pre-train | 0.6003 | 0.4838 | 9.5642 | 0.6209 | 0.8913 | 0.8144 | 7.8228 | 0.7880 | 0.4353 | 0.3024 | 47.9971 | 0.3619 |
| **Ours** | **0.7627** | **0.6544** | **3.5148** | **0.8197** | **0.9727** | **0.9471** | **1.6240** | **0.9742** | **0.7039** | **0.5706** | **14.9232** | **0.6196** |

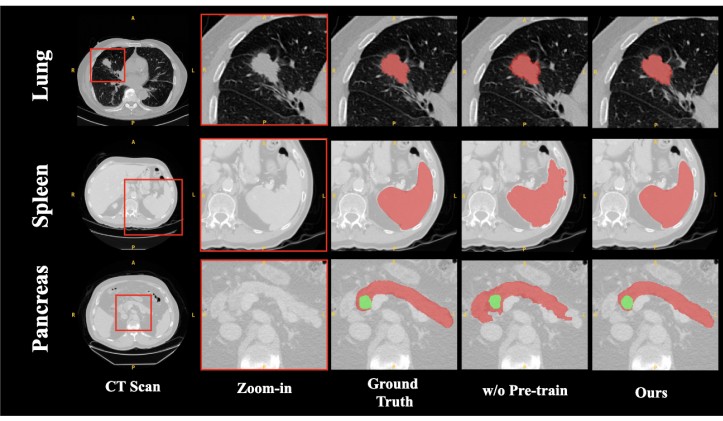

Figure 4: Visual comparison of segmentation results with and without Pre-training.

## D.2 EFFECTIVENESS OF SRPP MODULE

To validate the effectiveness of our proposed SRPP module, we compared our full model against an architectural variant where the SRPP module was removed ("w/o SRPP"). The quantitative results of this comparison are presented in Table 4.

The results demonstrate the effectiveness of the SRPP module in enhancing segmentation performance. Across all three datasets, the inclusion of SRPP brings consistent improvements. The impact is particularly pronounced on the more challenging datasets and in boundary-sensitive metrics. For instance, on the Pancreas dataset, integrating the SRPP module boosts the Dice score from 0.6459 to 0.7039 and significantly reduces the HD95 from 21.9181 to 14.9232. A similar substantial improvement in boundary delineation is observed for Lung, where the HD95 metric drops dramatically from 8.5837 to 3.5148.

This substantial improvement highlights the SRPP module's crucial role in capturing multi-scale contextual information. The visualization results in Fig. 5 further corroborate these findings.

Table 4: Ablation study of SRPP module on the three datasets. Best results are highlighted in **bold**.

| Method | Lung | | | | Spleen | | | | Pancreas | | | |
|---|---|---|---|---|---|---|---|---|---|---|---|---|
| | Dice ↑ | IoU ↑ | HD95 ↓ | NSD ↑ | Dice ↑ | IoU ↑ | HD95 ↓ | NSD ↑ | Dice ↑ | IoU ↑ | HD95 ↓ | NSD ↑ |
| w/o SRPP | 0.7535 | 0.6408 | 8.5837 | 0.8021 | 0.9672 | 0.9371 | 2.1951 | 0.9589 | 0.6459 | 0.5106 | 21.9181 | 0.5417 |
| **Ours** | **0.7627** | **0.6544** | **3.5148** | **0.8197** | **0.9727** | **0.9471** | **1.6240** | **0.9742** | **0.7039** | **0.5706** | **14.9232** | **0.6196** |

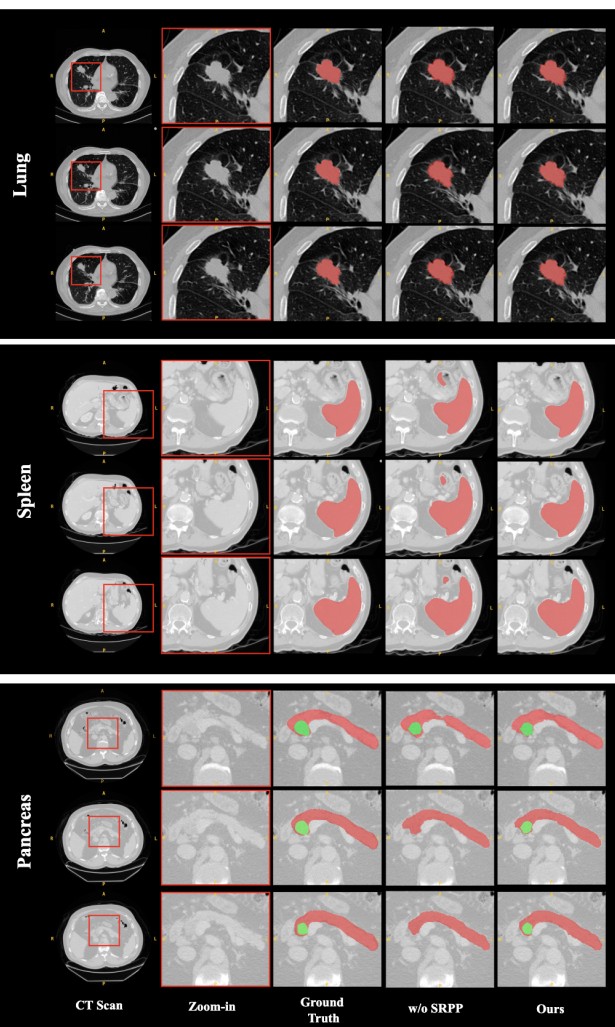

Figure 5: Visual comparison of segmentation results with and without SRPP module.

## D.3 EFFECTIVENESS OF BD MODULE

To validate the effectiveness of our proposed BD module, we designed an experiment comparing our full model with an ablated version that excludes this component ("w/o BD"). The quantitative results, presented in Table 5, highlight the consistent and positive impact of the BD module across all three medical imaging datasets. The most salient improvement is observed in the boundary-sensitive HD95 metric, particularly on the Lung dataset, where it drops dramatically from 9.8799 to 3.5148. Similarly, notable gains in Dice score are achieved on both the Lung (0.7366 to 0.7627) and Pancreas (0.6727 to 0.7039) datasets.

These empirical results and visualization in Fig. 6 strongly validate the design of our BD module. The significant reduction in Hausdorff Distance across the board confirms the module's efficacy in

guiding the network to learn more precise and accurate representations of organ boundaries. By explicitly focusing on the boundary details, our model is able to produce segmentation masks with higher fidelity, which is crucial for downstream clinical applications. Therefore, the BD module serves as an essential component for refining the segmentation results and improving overall performance.

Table 5: Ablation study of BD module on the three datasets. Best results are highlighted in **bold**.

| Method | Lung | | | | Spleen | | | | Pancreas | | | |
|--------|------|------|------|------|--------|------|------|------|----------|------|------|------|
| | Dice ↑ | IoU ↑ | HD95 ↓ | NSD ↑ | Dice ↑ | IoU ↑ | HD95 ↓ | NSD ↑ | Dice ↑ | IoU ↑ | HD95 ↓ | NSD ↑ |
| w/o BD | 0.7366 | 0.6186 | 9.8799 | 0.7836 | 0.9678 | 0.9379 | 1.9067 | 0.9587 | 0.6727 | 0.5406 | 18.7106 | 0.5897 |
| **Ours** | **0.7627** | **0.6544** | **3.5148** | **0.8197** | **0.9727** | **0.9471** | **1.6240** | **0.9742** | **0.7039** | **0.5706** | **14.9232** | **0.6196** |

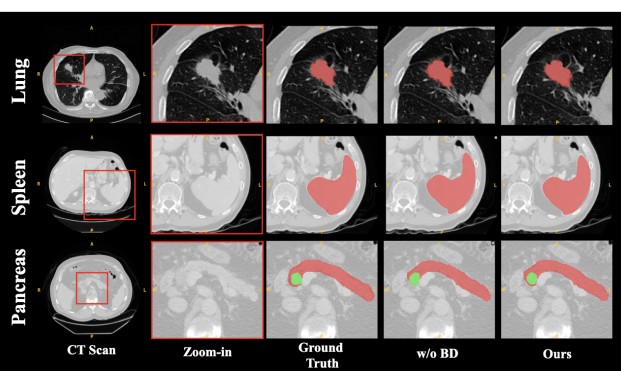

Figure 6: Visual comparison of segmentation results with and without BD module.

## D.4 EFFECTIVENESS OF BD-FUSION OPERATION

Finally, we investigate the effectiveness of our proposed fusion operation. This operation is designed to enhance the model's global boundary perception by fusing the initial input features of the main encoder with the high-level boundary features extracted by the BD module. We compare our full model against a variant where this fusion operation is omitted ("w/o BD-Fusion"), with the results detailed in Table 6 and visualization in Fig. 7. The data clearly indicates that removing this fusion step leads to a noticeable degradation in performance across all datasets.

Table 6: Ablation study of BD fusion operation on the three datasets. Best results are highlighted in **bold**.

| Method | Lung | | | | Spleen | | | | Pancreas | | | |
|--------|------|------|------|------|--------|------|------|------|----------|------|------|------|
| | Dice ↑ | IoU ↑ | HD95 ↓ | NSD ↑ | Dice ↑ | IoU ↑ | HD95 ↓ | NSD ↑ | Dice ↑ | IoU ↑ | HD95 ↓ | NSD ↑ |
| w/o BD-Fusion | 0.7346 | 0.6168 | 4.6238 | 0.7907 | 0.9697 | 0.9414 | 1.7239 | 0.9696 | 0.6670 | 0.5348 | 22.7462 | 0.5756 |
| **Ours** | **0.7627** | **0.6544** | **3.5148** | **0.8197** | **0.9727** | **0.9471** | **1.6240** | **0.9742** | **0.7039** | **0.5706** | **14.9232** | **0.6196** |

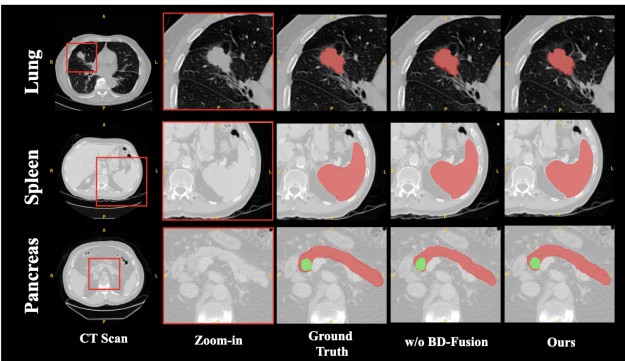

Figure 7: Visual comparison of segmentation results with and without BD-Fusion operation.

# E  ADDITIONAL EXPERIEMENTS

## E.1  IMPACT OF INPUT SLICE NUMBER

We conduct an experiment to investigate the impact of the number of input slices, a key hyperparameter that determines the amount of 3D contextual information provided to our model. We evaluate three different settings: 3, 6, and 12 slices. The experiments are performed on all three datasets, and the quantitative results are presented in Table 7.

The results show a clear and consistent trend: increasing the number of input slices generally leads to better segmentation performance across all datasets and most metrics. For instance, on the Pancreas dataset, increasing the slice number from 3 to 12 boosts the Dice score substantially from 0.5685 to 0.7039. Similarly, on the Spleen dataset, the Dice score improves from 0.9489 to 0.9727. This suggests that providing more spatial context helps the model achieve more accurate and robust segmentation.

Table 7: Experiments of different slice numbers on the three datasets. Best results are highlighted in **bold**.

| Method | Lung | | | | Spleen | | | | Pancreas | | | |
|---|---|---|---|---|---|---|---|---|---|---|---|---|
| | Dice ↑ | IoU ↑ | HD95 ↓ | NSD ↑ | Dice ↑ | IoU ↑ | HD95 ↓ | NSD ↑ | Dice ↑ | IoU ↑ | HD95 ↓ | NSD ↑ |
| Ours(slice 3) | 0.7457 | 0.6301 | 5.5617 | 0.7767 | 0.9489 | 0.9093 | 5.0349 | 0.9327 | 0.5685 | 0.4324 | 28.1533 | 0.4895 |
| Ours(slice 6) | 0.7627 | 0.6544 | **3.5148** | 0.8197 | 0.9700 | 0.9421 | 1.8402 | 0.9636 | 0.6645 | 0.5370 | 18.8110 | 0.5813 |
| Ours(slice 12) | **0.7643** | **0.6547** | 9.6687 | **0.8222** | **0.9727** | **0.9471** | **1.6240** | **0.9742** | **0.7039** | **0.5706** | **14.9232** | **0.6196** |

## E.2  IMPACT OF LOSS HYPERPARAMETERS

In addition to the number of input slices, we analyze the sensitivity of our model to two other crucial hyperparameters: relative spatial position prediction parameter $\lambda_1$ and boundary loss parameter $\lambda_2$. These parameters are critical as they control the final loss of the balance between relative self-supervision and boundary-based supervision. We conduct this analysis on three datasets, and the results are summarized in Table 8.

Our findings reveal a trade-off between different evaluation metrics. The setting with $\lambda_1$=0.01 and $\lambda_2$=0.3 yields the highest Dice (0.7636) and IoU (0.6550) scores, indicating the best overall region overlap. However, the setting with $\lambda_1$=0.01 and $\lambda_2$=0.1 achieves a significantly better (lower) HD95 score of 3.5148 and the highest NSD score of 0.8197. Since HD95 and NSD are more sensitive to the accuracy of segmentation boundaries, which is often paramount in medical applications, we prioritize performance on these metrics. Given that its Dice and IoU scores are still highly competitive and only marginally lower than the best, we conclude that the setting ($\lambda_1$=0.01 and $\lambda_2$=0.1) offers the best overall balance.

Therefore, we select bd=0.1 and rpp=0.01 as the default configuration for all main experiments presented in this paper.

Table 8: Analysis of different parameter settings on the Lung dataset. Best results are highlighted in **bold**.

| Method | Lung | | | |
|---|---|---|---|---|
| | Dice ↑ | IoU ↑ | HD95 ↓ | NSD ↑ |
| $\lambda_1$=0.01, $\lambda_2$=0.3 | **0.7636** | **0.6550** | 8.6419 | 0.8146 |
| $\lambda_1$=0.01, $\lambda_2$=0.5 | 0.7455 | 0.6335 | 3.7062 | 0.7997 |
| $\lambda_1$=0.03, $\lambda_2$=0.1 | 0.7506 | 0.6375 | 12.8412 | 0.7996 |
| $\lambda_1$=0.05, $\lambda_2$=0.1 | 0.7378 | 0.6260 | 6.3344 | 0.7882 |
| $\lambda_1$=0.01, $\lambda_2$=0.1 | 0.7627 | 0.6544 | **3.5148** | **0.8197** |

### E.3 IMPACT OF SRPP MEMORY

Given the effectiveness of the memory mechanism in our SAM-based segmentation and BD modules, a natural extension is to investigate whether our SRPP module could also benefit from a similar memory component. To this end, we conducted an experiment where we integrated a memory module between the SRPP's encoder and predictor, aiming to enhance its feature representation with historical context.

The results of this ablation study are presented in Table 9. Contrary to our initial hypothesis, the inclusion of the SRPP memory module does not lead to a consistent performance improvement. In fact, on the Spleen and Pancreas datasets, the model without this additional memory (Ours w/o SRPP memory) achieves significantly better results across most metrics. For instance, on the Spleen dataset, the Dice score drops from 0.9727 to 0.9554 when the memory is added. While the memory-equipped version shows a marginal gain in Dice/IoU on the Lung dataset, it comes at the cost of a substantially worse (higher) HD95 score.

This empirical evidence suggests that, for the SRPP module, adding a memory component may introduce redundant information or unnecessary architectural complexity that hinders the learning process, unlike in our other modules where it proved beneficial. This finding validates our final design choice. Therefore, to ensure optimal performance and maintain model efficiency, we do not include the memory module in the SRPP component in our final architecture.

Table 9: Experiments of the SRPP memory module. Best results are highlighted in **bold**.

| Method | Lung | | | | Spleen | | | | Pancreas | | | |
|---|---|---|---|---|---|---|---|---|---|---|---|---|
| | Dice ↑ | IoU ↑ | HD95 ↓ | NSD ↑ | Dice ↑ | IoU ↑ | HD95 ↓ | NSD ↑ | Dice ↑ | IoU ↑ | HD95 ↓ | NSD ↑ |
| Ours w/ SRPP memory | **0.7635** | **0.6551** | 10.9104 | 0.8180 | 0.9554 | 0.9198 | 5.4293 | 0.9397 | 0.6969 | 0.5524 | **12.7863** | 0.5780 |
| Ours w/o SRPP memory | 0.7627 | 0.6544 | **3.5148** | **0.8197** | **0.9727** | **0.9471** | **1.6240** | **0.9742** | **0.7039** | **0.5706** | 14.9232 | **0.6196** |

## F    MORE VISUALIZATION

In this section, we present additional 3D renderings of our segmentation results to provide a more holistic and intuitive assessment of our model's performance. The following figures showcase representative examples from three datasets.

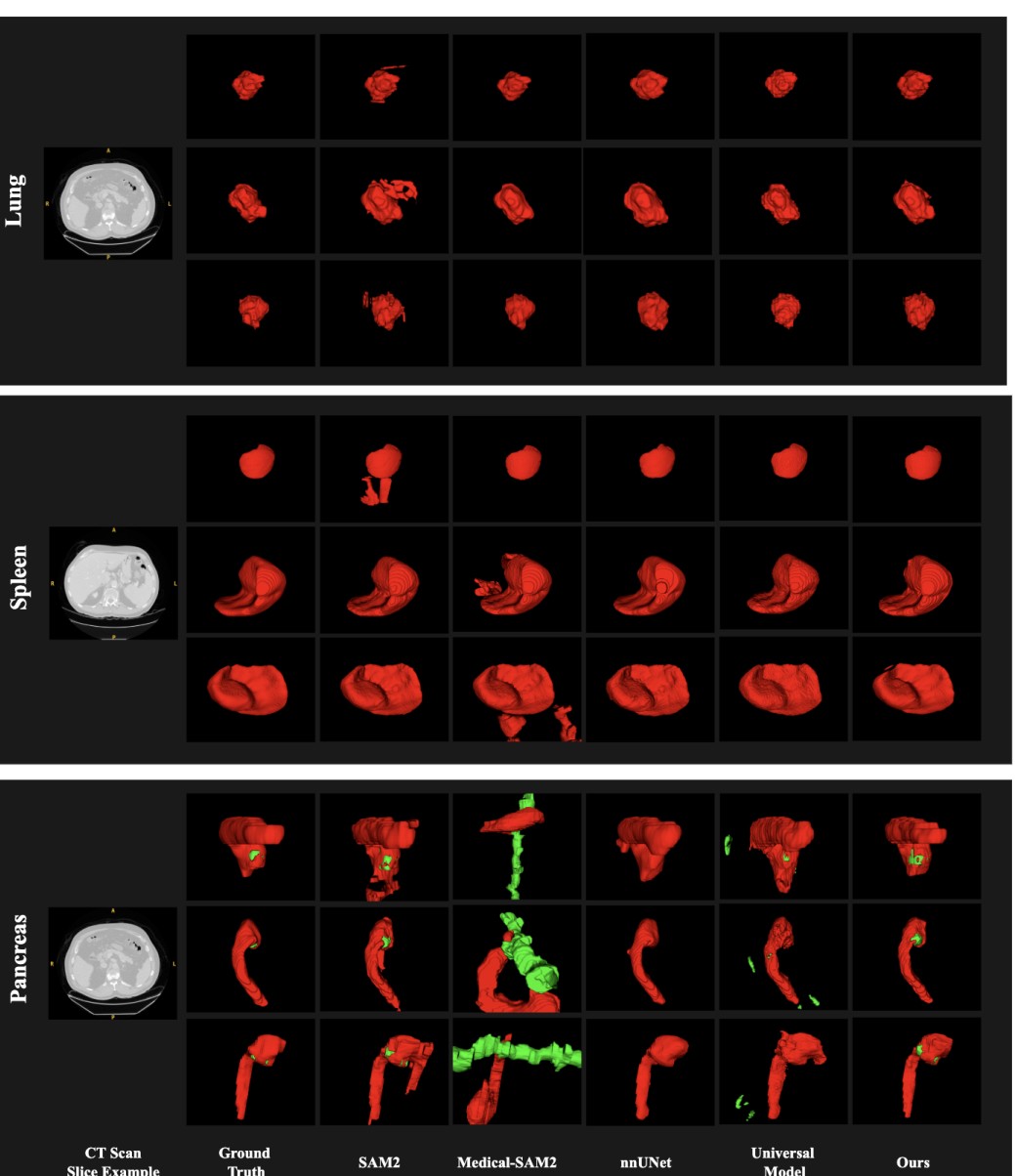

Figure 8:   Three-dimensional renderings of segmentation results on representative test cases. From top to bottom: examples from the Lung, Spleen, and Pancreas datasets. Each visualization displays the model's prediction 3D model.

