# OpenReview forum: "SAM2-3dMed: Empowering SAM2 for 3D Medical Image Segmentation"
_ICLR.cc/2026/Conference — Submitted to ICLR 2026_

### Official Review · Reviewer_Hun1 · 2025-10-25

**Soundness:** 3
**Presentation:** 2
**Contribution:** 2
**Rating:** 2
**Confidence:** 4

**Summary:**

The goal is to transfer the temporal memory of SAM2 to the slice dimension of volume data, improving cross-slice consistency and boundary accuracy in 3D medical segmentation. To this end, we propose two components: SRPP and BD .

**Strengths:**

1. A modular design with clear principles and low annotation costs.
The key challenges in migrating the SAM2 video model to volumetric data are broken down into two key areas: cross-slice spatial dependency (SRPP, self-supervision, no need for new annotations) and boundary accuracy (BD, parallel branching, explicit boundary supervision). These two modules each perform their respective functions and complement each other, creating a closed loop between motivation, method, and loss function.

2. Engineering-friendly and budget-friendly.
During training, the SAM2 image encoder is frozen, and only the memory attention and decoding/new branches are fine-tuned. A unified mini-batch and standard optimizer/scheduling strategy facilitates reproducibility and implementation, and reduces the risk of overfitting in settings with limited medical data.

3. Comprehensive Experimental Coverage and Effective Ablation Decisions
This experiment covers three representative MSD tasks (lung tumors, spleen, and pancreas/tumors) and presents multiple metrics (Dice, Intersection of Union (IoU), HD95, and Non-Sensitive Distance (NSD)). It also conducts ablation tests on key designs, such as verifying that adding memory to SRPP can lead to instability. This led to a more concise and effective final architecture, demonstrating empirically driven design choices.

**Weaknesses:**

1.Insufficient demonstration of novelty and necessity.
SRPP is essentially a self-supervised task of "inter-slice relative position regression/ranking," closely resembling common temporal/sequential prediction in videos. However, the paper lacks a systematic comparison with existing approaches for "spatial videoization" or 3D context modeling (3D convolution, cross-slice attention, learnable position encodings), and its rationale for the necessity of SRPP is weak. The authors primarily focus on conceptual explanation (Figure 1 + Methods section), lacking empirical comparisons with alternative designs.


2. The SRPP target definition may not match the characteristics of medical data.
Using "j-i" as the relative position ground truth assumes consistent slice thickness and orientation across all volumes. However, medical CT/MRI often exhibits anisotropy and varying slice thicknesses, and even discrepancies in preprocessing such as resampling and flipping. No reports have been found on whether physical coordinate/spacing normalization or orientation standardization is performed, raising questions about whether the "distances" learned by SRPP have consistent physical meaning across cases.

3. The value-cost trade-off of the BD branch is unclear. The BD branch operates in parallel with the mask decoder, but the structure and parameter sharing with the main decoder, the additional FLOPs and memory overhead, and whether it is retained during inference remain unquantified. For clinical implementation, the trade-off between additional complexity and benefits requires clearer data.

4. Evaluation Protocol and Comparability Issues. The datasets were all split using the authors' own "80% training set/20% test set" (derived from the MSD training set). No official test server or cross-validation was used. This can easily overestimate generalization and poorly compare with the literature. At a minimum, the mean ± variance of multiple random splits should be reported. Does the Pancreas task evaluate multiple categories of "organs + tumors"? Table 1 lists "K categories", but no per-category or average definition (macro/micro) is found. It is unclear whether the label definition is consistent with baselines such as nnUNet.

5. Style and Formatting: The section title "EXPERIEMENTS" is misspelled; some sentences are lengthy or repetitive.

6. The loss weights are inconsistently expressed: the text defines λ1/λ2, but the appendix conclusion uses "bd=0.1, rpp=0.01". The symbols need to be unified, and the optimization method (grid/Bayesian/fixed) needs to be clarified.

**Questions:**

1. SRPP and physical scale: Should physical coordinates (mm) be used in SRPP instead of slice indices? How to maintain the correctness of "relative position" supervision if the data is resampled/flipped?

2. Metrics settings: τ value and units for NSD? What are the units for HD95? Are all methods uniformly resampled to the same voxel spacing before calculation?

3. Fair comparison: Do all baselines use the same data split, preprocessing and data augmentation, and training budget (epoch/early stopping/learning rate schedule)? Is the nnUNet "self-configuration" process fully followed?

4. Computational cost: throughput, memory, and latency for training and inference; how does the cost compare to the alternative of "backbone-only + 3D position encoding/cross-slice attention"?

---

### Official Review · Reviewer_URgc · 2025-10-31

**Soundness:** 2
**Presentation:** 2
**Contribution:** 2
**Rating:** 2
**Confidence:** 4

**Summary:**

This paper presents SAM2-3dMed, a novel framework for adapting the video foundation model SAM2 to 3D medical image segmentation. The authors identify two key domain gaps: 1) the mismatch between SAM2's unidirectional temporal modeling (for videos) and the bidirectional spatial continuity of anatomical volumes , and 2) the lack of focus on precise boundary delineation in video tasks compared to clinical requirements. To address this, the proposed method introduces two key innovations while keeping the SAM2 image encoder frozen : SRPP module, and a parallel BD module with an auxiliary loss to enhance contour accuracy. Experiments on three MSD datasets demonstrate that SAM2-3dMed achieves superior performance in both overlap and boundary metrics compared to several baselines.

**Strengths:**

- The paper is well-motivated for addressing two fundamental domain gaps in models like SAM2.

- The two proposed modules are well-designed and address the identified weaknesses.

- The method is compared against a strong suite of baselines, including the standard nnUNet, other foundation-model adaptations like MedSAM-2, and a directly fine-tuned SAM2.

**Weaknesses:**

- There is the inconsistently used hyperparameters for the main results. A close comparison of Table 1 and Table 7 reveals that the "Ours" results are not from a single model. The Lung results are taken from the 6-slice configuration. In contrast, the Spleen and Pancreas results are from the 12-slice configuration. Presenting results from different model configurations under a single "Ours" entry in Table 1 is somewhat misleading.

- Some visualizations in Figure 3 are marginal. If the authors would like to highlight frame consistency and better boundary results, it would be better to emphasize this in the text and show contours in the figure.

- The experimental validation is limited, comprising only three tasks. Furthermore, the datasets for these tasks are relatively small, ranging from 41-281 cases, which may raise concerns about the generalizability of the results.

- The paper is lacking an efficiency analysis. The authors should report computational costs (e.g., parameters, inference time) for their proposed model, especially for the 12-slice configuration which yielded the best results for the Spleen and Pancreas datasets, to assess its practical viability.

- The author’s claim in Figure 1(c) that the requirement for boundary delineation in video segmentation is low might be too strong.

**Questions:**

Please see Weakness part.

---

### Official Review · Reviewer_NgR1 · 2025-11-01

**Soundness:** 3
**Presentation:** 3
**Contribution:** 3
**Rating:** 6
**Confidence:** 4

**Summary:**

The paper introduces $\text{SAM2-3DMED}$, a novel framework designed to adapt the powerful video object segmentation model, Segment Anything Model 2 ($\text{SAM2}$), for interactive $3\text{D}$ medical image segmentation. The core motivation is to bridge two critical domain gaps: $1)$ the difference between $\text{SAM2}$'s native $\text{unidirectional temporal flow}$ processing and the $\text{bidirectional anatomical continuity}$ inherent in volumetric medical scans, and $2)$ the need for highly precise boundary delineation in clinical settings ($\text{HD95} \downarrow$ metric importance).

**Strengths:**

The paper exhibits high originality by being one of the first works to successfully and systematically adapt the $\text{SAM2}$ architecture for $3\text{D}$ medical image segmentation. Simply applying $\text{SAM2}$ directly to $3\text{D}$ data is non-trivial due to the domain gap, and the authors propose a creative, targeted solution. The Slice Relationship Prediction and Propagation ($\text{SRPP}$) Module is a particularly novel component, as it formalizes the slice-to-slice continuity problem as a structured propagation mechanism, going beyond simple $3\text{D}$ convolution or basic transformer blocks.

**Weaknesses:**

While the paper focuses heavily on architectural improvements, a major aspect of SAM-style models is their prompt robustness. The authors use fixed, center-based $3\text{D}$ points for interactive prompts. This is a simplification. A major weakness is the lack of analysis on:
- Varying Prompt Quality: How does $\text{SAM2-3DMED}$ perform when given noisy, sparse, or ill-placed prompts, which are common in real clinical interaction?
- Comparison of Prompt Types: A comparison with box prompts or scribbles, especially across slices, would better validate the interactive capability for $3\text{D}$ volumes.
The $\text{SAM2}$ backbone is inherently heavy. While the paper touts improved performance, there is a missing discussion on the computational burden relative to existing lightweight $3\text{D}$ segmentation models (e.g., simpler $3\text{D}$ U-Net variants). For clinical deployment, inference time and memory footprint (especially for high-resolution $3\text{D}$ volumes) are crucial. Providing a comparison of inference speed (frames per second or volume processing time) would significantly strengthen the argument for its practical utility.

**Questions:**

- Does the Slice Relationship Prediction component inherently struggle or perform differently when segmenting multiple distinct organs/targets within the same $3\text{D}$ volume, or when dealing with highly sparse targets? Since the slice relationships are predicted, I'm curious if the presence of many small, separate objects negatively impacts the quality of the predicted feature propagation path. Please provide any analysis or intuition on multi-target segmentation performance.
- Can the authors provide visual examples of where $\text{SAM2-3DMED}$ struggles (e.g., highly thin structures, boundaries near low-contrast tissues, or small pathological lesions)? This would provide crucial insight into the remaining limitations of the $3\text{D}$ adaptation strategy.
- I strongly recommend including a table comparing the Inference Speed (e.g., total processing time per $3\text{D}$ volume in seconds) of $\text{SAM2-3DMED}$ against at least two existing state-of-the-art $3\text{D}$ medical segmentation models (e.g., a strong $3\text{D}$ U-Net variant and a lightweight $3\text{D}$ SAM adaptation). This metric is critical for assessing the clinical viability of the model.

---

### Official Review · Reviewer_uGXL · 2025-11-03

**Soundness:** 3
**Presentation:** 3
**Contribution:** 2
**Rating:** 2
**Confidence:** 5

**Summary:**

This argues that SAM2 isn’t ideal for 3D medical segmentation because SAM2 might rely on a unidirectional time based progression of objects in a video stream. But a 3D biomedical volume might not have much temporal properties like a video stream. Therefore, they introduce a 3D slice relative position module which helps the model learn about the 3D structure of the biomedical volume independent of the time-progression. They also have a module to better predict boundaries and a segmentation loss. The results show that these modules do infact help get better 3D segmentation results on biomedical data.

**Strengths:**

Thorough experiments on a reasonable range of datasets and ablation results showing the efficacy of their approach.

**Weaknesses:**

- So there has been a flood of bio-medical 3D segmentation tools based on SAM,SAM2 recently, and this feels like ‘another SAM-based segmentation’ paper.

- I really like the hypothesis that SAM2’s temporal unidirectional predisposition might not be suitable for biomedical applications. However, the authors went about the wrong way to test and prove this. Showing final segmentation tells me that this hypothesis was useful in improving results, but doesn’t convince me that this temporal bias does in fact exist, and using the slice position prediction resolves it. Please read this paper https://arxiv.org/pdf/2101.01600 for ideas. Fig. 6 in the paper shows how the relative positions of the features change as we predict closer and closer in the future in a given direction. Please note that I am not asking the authors to convert their work to a hyperbolic learning paper. What I am asking them to do is use a visualization technique like in the paper above to show that prediction forward and backward in time gives different properties at the representation level, which also reveal themselves on biomedical volumes as well. And their relative location module helps fix that.

- Similarly, this other hyperbolic segmentation paper (https://arxiv.org/pdf/2203.05898 ) shows that segmentation predictions at boundaries have a higher uncertainty. And the authors need to show, in whatever way they can, that their boundary prediction module helps reduce uncertainty at the boundary pixels.

- Bringing ideas like above will help convey a more well-rounded and convincing story.

**Questions:**

None.

---

### Meta-Review · Area_Chair_jiCe · 2026-01-03

**Summary:**

This paper introduces SAM2-3dMed that adapts the foundation model SAM2 for 3D medical image segmentation. It addresses two gaps: the mismatch between SAM2’s unidirectional temporal modeling and the spatial continuity of 3D volumes, and the insufficient boundary precision required for clinical tasks. Experiments on MSD datasets show consistent improvements in both overlap and boundary metrics. The authors do not respond to the reviewers’ comments, and the three reviewers have the negative scores.

**Reviewer Concerns:**

The authors do not respond to the reviewers’ comments. Therefore, I think that the reviewers’ concerns have been not addressed.

**Reviewer Scores:**

I believe that none of the four reviewers will change their scores due to no response.

---

### Decision · Program_Chairs · 2026-01-26

Reject